Larger hydrological simulation uncertainties where runoff generation capacity is high: insights from 63 catchments in southeastern China Yi Nan<sup>1\*</sup>, Fuqiang Tian<sup>1,2</sup>, Mahmut Tudaji<sup>1,2</sup> 1. State Key Laboratory of Hydroscience and Engineering, Tsinghua University, Beijing 100084, China 2. Department of Hydraulic Engineering, Tsinghua University, Beijing 100084, China

11 Correspondance: Yi Nan (ny1209@qq.com)

## Abstract

Traditional parameter calibration strategies that focus on a single optimal parameter set may lead to large uncertainties and biases in simulating internal hydrological processes because of parameter equifinality. This study used the semi-distributed Tsinghua Hydrological Model based on Representative Elementary Watershed (THREW) to investigate the influence of parameter equifinality on uncertainties in surface–subsurface runoff partitioning. The model was implemented in 63 catchments in southeastern China with high-quality rainfall and streamflow data. Behavioral parameter sets were selected based on KGE thresholds to quantify uncertainty in estimates of the contribution of subsurface runoff (C<sub>sub</sub>). Correlation analyses were conducted to investigate factors influencing these uncertainties. Results showed that: (1) the THREW model performed well across the 63 catchments, with an average optimal KGE (KGE<sub>opt</sub>) of 0.846. C<sub>sub</sub> varied widely among catchments, ranging from 1.0% to 74.1% (mean = 31.7%), and was below 50% in 84% of the catchments, indicating that surface runoff was the dominant runoff generation mechanism in the study area. (2) Substantial uncertainty in C<sub>sub</sub> can arise from small differences in KGE, with notable variability among catchments. The uncertainty in C<sub>sub</sub> was modest in most catchments, with mean Bias (difference between the

28 C<sub>sub</sub> estimated using the optimal set and the average across all behavioral parameter sets) and 29 Range (max-min across behavioral sets) of 2.7% and 15.8%, respectively. However, the 30 uncertainty can be large in some catchments, where reliance on a single optimal parameter set 31 is likely inappropriate. (3) Runoff ratio was identified as an important catchment attribute 32 significantly correlated with C<sub>sub</sub> and its uncertainty. In catchments with stronger runoff-33 generation capacity, the model tended to be less sensitive and the simulation of internal runoff-34 component partitioning tended to exhibit larger uncertainties. Such evidence can provide 35 empirical a priori guidance on the likely magnitude of uncertainties and help inform calibration-36 strategy selection.

#### 1. Introduction

Hydrological models are useful tools for understanding hydrological processes and predicting variability in water resources. Parameter equifinality is a widely recognized issue in hydrological modelling, meaning that different parameter sets may yield similar model performance (Gupta et al., 2008). This can result in large uncertainties in the simulation of internal hydrological processes despite producing similar total hydrographs (Delavau et al., 2017; Nan & Tian, 2024). Understanding the characteristics and mechanisms of uncertainty generation and propagation, systematically assessing uncertainty, and employing multiple approaches to reduce uncertainty are important tasks in the field of hydrological modelling. Parameter calibration is a necessary step in developing hydrological models. Streamflow is the most commonly used data type for calibration, and other variables—such as soil moisture, evapotranspiration, snow cover/depth, glacier mass balance, and isotopic tracers—are also employed by some researchers to better constrain uncertainties (Chen et al., 2017; He et al., 2019; Ala-aho et al., 2017; He & Pomeroy, 2023). Regardless of the datasets and objective functions used (e.g., Nash-Sutcliffe efficiency, NSE; Kling-Gupta efficiency, KGE), calibration strategies can be broadly classified into two types. One seeks a single optimal parameter set that maximizes the chosen performance metric, which is often assumed to represent realistic hydrological behavior and is subsequently used for process inference or

performance threshold to Monte Carlo realizations or to parameter sets sampled during 58 59 calibration (Blasone et al., 2008; Nan et al., 2021). 60 Ensemble-based strategies not only support investigation of hydrological processes but 61 also allow for a systematic assessment of model sensitivity and uncertainty. In practice, the 62 choice of calibration strategy often depends on research objectives: ensemble methods are more commonly applied in studies explicitly focused on uncertainties (He et al., 2019), while single-63 64 optimal-parameter approaches remain the default in many applications (Beven & Binley, 2013; Efstratiadis & Koutsoyiannis, 2010). From the perspective of model behavior, ensemble 65 approaches should be preferable in regions where parameter equifinality is pronounced, 66 67 whereas a single optimal parameter set may be adequate in catchments with limited equifinality. However, existing studies rarely provide an a priori assessment of the likely degree of 68 69 equifinality for a given catchment, so researchers lack practical guidance for selecting the most 70 appropriate calibration strategy. This prevailing preference for single-best solution has 71 constrained advances in understanding model uncertainty to some extent. 72 The contribution of runoff components to streamflow is an important hydrological 73 characteristic that reflects runoff-generation mechanisms and is susceptible to parameter 74 equifinality (He et al., 2021). Previous studies have shown that, in some catchments, the 75 partitioning between surface and subsurface runoff can exhibit large uncertainty even when 76 streamflow simulations are classified as behavioral. Analyzing uncertainties in runoff 77 components and their controlling factors is essential for deepening our understanding of runoff-78 generation processes and for improving the reliability of hydrological simulations and 79 predictions (Cui & Tian, 2025; Tudaji et al., 2025a). However, among studies addressing 80 uncertainties and sensitivities in hydrological models, the emphasis has primarily been on 81 parameter uncertainty, model overall performance and the simulation of total runoff (Di Marco 82 et al., 2021; Yang et al., 2019), whereas uncertainty in internal water partitioning has received 83 comparatively little attention. This lack of focus likely arises because internal water states and 84 fluxes are harder to observe than total runoff, making the validation of internal partitioning 85 more difficult and less common (Nan et al., 2025).

scenario analysis, for example to assess runoff response to climate change (Su et al., 2023). The

other approach employs an ensemble of parameter sets, typically obtained by applying a

Motivated by the above background, this study utilized a semi-distributed hydrological model to investigate the influence of parameter equifinality on internal water component partitioning. In specify, the objectives of this study are: (1) to quantify the uncertainty in the contribution of subsurface runoff ( $C_{sub}$ ) resulting from small changes in model performance metric (KGE), and (2) to identify factors that influence model uncertainty so as to provide potential a priori empirical guidance for assessing uncertainty and selecting calibration strategies.

## 2. Materials and methodologies

## 94 2.1 Catchment and data set

This study utilized the catchment set established by Tudaji et al. (2025b) to analyze the hydrological simulation uncertainties. It included 63 small- to medium-scale catchments located in southeastern China (Figure 1), most of which fall within the Yangtze River Basin. The catchment attributes vary considerably (Table 1). The drainage areas of the catchments range from 91.5 to 5,266 km², with an average of 1,528 km². These catchments exhibit significant diversity in climate, topography, and rainfall–runoff relationships, with wide ranges of mean annual rainfall (647–2,593 mm), topographic slope (1.8–26.2°), and runoff ratios (0.31–0.96).

**Figure 1.** Geographic distribution of the study catchments (Adapted from Tudaji et al, 2025b).

Table 1. Statistical summaries of the catchment attributes

| Attribute | Description          | Min              | Max  | Average | Unit |
|-----------|----------------------|------------------|------|---------|------|
| DRA       | Drainage area        | 92               | 5266 | 1528    | km²  |
| MAR       | Mean annual rainfall | 647              | 2593 | 1531    | mm   |
| MAQ       | Mean annual runoff   | 356              | 1571 | 868     | mm   |
| QR        | Runoff ratio         | 0.31             | 0.96 | 0.58    | -    |
| TS        | Topographic slope    | 1.83 26.18 11.77 |      | O       |      |

Hourly discharge and rainfall data from 1 January 2014 to 31 December 2015 were collected from the National Rainfall and Hydrological Database, established by the Information Center of the Ministry of Water Resources (http://xxfb.mwr.cn/sq\_dtcx.html, last access: 10 December 2023). Considering the variable quality of the raw data (mainly completeness and temporal resolution), 63 hydrological stations were selected based on high-quality standards. Specifically, the average temporal resolution—defined as the ratio of the time period to the number of measurements—exceeded 3,650 s. This threshold was slightly higher than 1 h, ensuring hourly resolution while allowing for individual missing data. In addition, to make full use of the water level data (which are generally more complete than discharge data) in the

118119

127128

database, the water level-discharge relationship was used to infer discharge values for periods with only water level data. To ensure the accuracy of these calculations, selected stations met the criteria that discharge data for more than 80% of the time steps also had water level data, and the coefficient of determination (R2) for the water level-discharge relationship exceeded 0.95. Similar to the discharge data, 863 rainfall stations with hourly records were selected within the spatial extent of the 63 catchments. The Thiessen polygon method (Han and Bray, 2006) was employed to generate areal rainfall for each catchment. Each catchment was covered by 15 Thiessen polygons on average. Other data used for model setup were collected from public datasets. The 90 m-resolution Multi-Error-Removed Improved-Terrain Digital Elevation Model (MERIT DEM; Yamazaki et al., 2017) was used for catchment extraction. Daily temperature and potential evapotranspiration were sourced from ERA5-Land (Muñoz Sabater, 2019). Soil parameters were estimated from the global high-resolution dataset of soil hydraulic and thermal parameters (Dai et al., 2019). The 8-day leaf area index (LAI) and 16-day normalized difference vegetation index (NDVI) products from MODIS (MOD15A2H, Myneni et al., 2021; MOD13A1, Didan,

### 2.2 THREW Hydrological model

2021) were used to represent vegetation conditions.

The Tsinghua Hydrological Model Based on Representative Elementary Watershed (THREW), developed by Tian et al. (2006), was employed in this study. The THREW model is based on a set of balance equations for mass, momentum, and energy, along with constitutive equations controlling fluxes among simulation units. Each catchment is first divided into several Representative Elementary Watersheds (REWs) based on DEM data, and several subzones are defined within each REW, serving as the elementary simulation units of the THREW model (Figure 2). Specifically, each REW is divided into surface and subsurface layers. Six sub-zones are defined in the surface layer based on underlying types: vegetation zone (v-zone), bare soil zone (b-zone), snow zone (n-zone), glacier zone (g-zone), sub-stream network zone (t-zone), and main channel reach zone (r-zone). Two sub-zones are defined in the subsurface layer according to soil saturation conditions: unsaturated zone (u-zone) and saturated zone (s-

zone). Considering the subtropical monsoon climate of the selected catchments—characterized by high mean temperatures and the temporal coincidence of peak rainfall and temperature—the influence of cryospheric processes on runoff is negligible; therefore, the snow and glacier modules and the corresponding sub-zones were not activated in this study. The THREW model has been described in our previous publications; for full model details see Tian et al. (2006). This study primarily investigates uncertainties in runoff-component partitioning using the THREW model.

Figure 2. Schematic illustration of the THREW model (Adapted from Tudaji et al., 2025b)

## 2.3 Model calibration

The parameters to be calibrated are listed in Table 2. The Python Surrogate Optimization Toolbox (pySOT; Eriksson et al., 2019) was employed for model calibration. The pySOT algorithm utilizes radial basis functions (RBFs) as surrogate models to approximate model simulations and reduce runtime per iteration. During each pySOT run, parameters were generated using the symmetric Latin hypercube design (SLHD), and the optimization stopped when the objective converged or the number of model runs reached a threshold (set to 3000 in

171172

this study). The Kling–Gupta efficiency (KGE) was used as the optimization objective to reflect overall model performance in terms of correlation, variability, and bias (Eq. 1).

$$KGE = 1 - \sqrt{(1-r)^2 + (1-\alpha)^2 + (1-\beta)^2}$$
 (1)

where, r represents the Pearson correlation coefficient between simulated and observed values,  $\alpha$  is the ratio of the mean of simulated values to that of observed values,  $\beta$  is the ratio of the standard deviation of simulated values to that of observed values.

 Table 2. Descriptions of calibrated parameters in the THREW model (Adapted from Tudaji et

al., 2025b)

| Symbol | Unit | Physical description                                           |     |  |  |  |  |
|--------|------|----------------------------------------------------------------|-----|--|--|--|--|
| WM     | cm   | Tension water storage capacity for saturation area calculation |     |  |  |  |  |
| В      | -    | Shape coefficient for saturation area calculation              |     |  |  |  |  |
| KKA    | -    | Exponential coefficient for groundwater outflow calculation    |     |  |  |  |  |
| KKD    | -    | Linear coefficient for groundwater outflow calculation         |     |  |  |  |  |
| C1     | -    | Coefficient for runoff concentration calculation using         | 0-1 |  |  |  |  |
|        |      | Muskingum method                                               |     |  |  |  |  |
| C2     | -    | Coefficient for runoff concentration calculation using         | 0-1 |  |  |  |  |
|        |      | Muskingum method                                               |     |  |  |  |  |

This study aimed to investigate model performance when the evaluation metric is relatively high. Consequently, for each catchment the pySOT algorithm was repeated 50 times, yielding 50 parameter sets. Parameter sets whose KGE exceeded a threshold were selected as behavioral parameters; the threshold was defined as 0.05 below the optimal KGE (KGE<sub>opt</sub>) obtained among the 50 sets.

### 2.4 Quantification of surface-subsurface runoff partitioning and its uncertainty

The surface–subsurface runoff partitioning in each catchment was analyzed based on THREW outputs of internal variables (Figure 2). Surface and subsurface runoff were defined as two runoff components based on runoff generation pathways as reviewed by He et al. (2021). Surface runoff includes rainfall occurring in areas where soils are saturated, where rainfall intensity exceeds infiltration capacity, or in impermeable areas such as river channels. Subsurface runoff is the outflow from the saturated zone. The contribution of subsurface runoff

to total runoff (C<sub>sub</sub>) was calculated to represent the surface–subsurface runoff partitioning characteristic (Eq. 2).

$$C_{sub} = \frac{q_{sub}}{q_{sur} + q_{sub}} \times 100\% \tag{2}$$

- where, Q<sub>sur</sub> and Q<sub>sub</sub> are the amount of surface and subsurface runoff, respectively, which can be obtained by the model outputs.
- The uncertainty of C<sub>sub</sub> was analyzed from two aspects. The first aspect is the 187 representativeness of C<sub>sub</sub> estimated by the optimal parameter set, quantified by bias and relative 188 bias (RBias) between the C<sub>sub</sub> estimated using the optimal set and the average C<sub>sub</sub> across all 189 behavioral parameter sets (Eqs. 3 and 4). The second aspect is the variability of C<sub>sub</sub> across the 190 behavioral parameter sets, quantified by the standard deviation (STD) and the range of C<sub>sub</sub> 191 among those sets (Eqs. 5 and 6). To ensure adequate parameter set samples for uncertainty 192 analysis, only the catchments where the number of behavioral parameter sets exceed 10 were 193 selected for this analysis.

$$Bias = \left| C_{sub}^{opt} - \overline{C_{sub}} \right| \tag{3}$$

$$RBias = \frac{Bias}{\overline{C_{sub}}} \times 100\% \tag{4}$$

$$STD = \sqrt{\frac{\sum_{i=1}^{n} (c_{sub,i} - \overline{c_{sub}})^2}{n}}$$
 (5)

$$Range = C_{sub}^{max} - C_{sub}^{min}$$
 (6)

where,  $C_{sub}^{opt}$  is the  $C_{sub}$  estimated by the optimal parameter set,  $\overline{C_{sub}}$  is the average  $C_{sub}$  estimated by the behavioral parameter sets, n is the number of behavioral parameter sets,  $C_{sub}^{max}$  and  $C_{sub}^{min}$  are the maximum and minimum of  $C_{sub}$  estimated by the behavioral parameter sets.

#### **201 3. Results**

## 3.1 Model performance

Among the 63 catchments, 50 had more than 10 behavioral parameter sets producing KGE higher than KGE<sub>opt</sub>–0.05 (Figure 3a). The KGE<sub>opt</sub> in these 50 catchments ranged from 0.663 to 0.947, with an average of 0.846, and 30 catchments achieved KGE<sub>opt</sub> values above 0.85. The comparisons of simulated and observed hourly streamflow in two typical catchments (with the

210211

highest and lowest KGE<sub>opt</sub>) are presented in Figure 4, showing strong consistency between model outputs and observations. These metrics and results indicate generally good model performance in the study area.

**Figure 3.** Summary of catchment counts with different (a) optimal KGE, (b) numbers of behavioral parameter sets, and (c) contributions of subsurface runoff

Figure 4. Comparison of simulated and observed hourly discharge in two typical catchments

with the highest and lowest KGE: (a) Xintian, (b) Yongshun

The number of behavioral parameter sets varied significantly among these 50 catchments (Figure 3b). The average number was 24.28, and in 15 catchments this number fell in the range 11–15. In only one catchment were all 50 parameter sets obtained by the pySOT process identified as behavioral. Considering the random generation of initial parameter sets within each pySOT running, the number of behavioral parameter sets serve as a partial indicator of model sensitivity.

# 3.2 Surface-subsurface runoff partitioning and its uncertainties

For each catchment, the average  $C_{sub}$  estimated across all behavioral parameter sets was taken as the final estimation. As shown in Figure 3c,  $C_{sub}$  varied markedly among catchments, ranging from 1.0% to 74.1% with an average of 31.7%. In 42 of the 50 catchments,  $C_{sub}$  was below 50%, indicating that surface runoff was the dominant runoff generation mechanism. This dominance can be attributed to the generally wet climate in the study area, which leads to high soil water saturation and extensive saturated areas.

Figure 5 illustrates the uncertainty of  $C_{sub}$  in two typical kinds of catchments. In the Heishui catchment (Figure 5a),  $C_{sub}$  exhibited strong variability when KGE was within 0.05 of its optimal value. The bias between the  $C_{sub}$  when the optimal KGE achieved ( $C_{sub}^{opt}$ ) and the

average  $C_{sub}$  produced by the behavioral parameter sets ( $\overline{C_{sub}}$ ) was 13%, accounting for 23% of  $\overline{C_{sub}}$ . Moreover, the  $C_{sub}$  estimated by each individual behavioral parameter set ranged from 32% to 85%, and the difference between the maximum and minimum  $C_{sub}$  was as high as 53%, indicating that different behavioral parameter sets may imply different dominant runoff generation mechanisms. By contrast, in the Shuikou catchment (Figure 5b), although  $C_{sub}$  estimated by the original 50 parameter sets also spanned a wide range, the model performance was much more sensitive to  $C_{sub}$  than in the Heishui catchment, as evidenced by a sharp decline in KGE when  $C_{sub}$  deviated from  $C_{sub}^{opt}$ . As a result, only 13 parameter sets produced KGE higher than the behavioral threshold, yielding small values of Bias (0.6%) and Range (6%).

Figure 5. Relation between KGE and contribution of subsurface runoff in two typical catchments: (a) Heishui, (b) Shuikou. Optimal parameter set (red triangle), ensemble mean (blue line), and range (gray shading) are shown, and associated uncertainty metrics are illustrated.

Figure 6 shows the boxplot of C<sub>sub</sub> uncertainty metrics derived by behavioral parameter sets in 50 catchments. Regarding the representativeness of the optimal C<sub>sub</sub>, the Bias metric ranged from 0.1% to 19.8%, with an average of 2.7%. Bias was less than 3% in 40 of the 50 catchments, indicating that the C<sub>sub</sub> obtained from the optimal parameter set was close to the mean across all behavioral parameter sets. However, Bias exceeded 10% in 3 catchments, indicating that relying solely on the optimal parameter set may still lead to misestimation of surface–subsurface runoff partitioning. The RBias metric exhibited a wider range, reaching a maximum of 75.5%, likely due to variations in average C<sub>sub</sub> (i.e., the denominator in Eq. 2). As for the variability of the behavioral results, the STD and Range of C<sub>sub</sub> varied from 0.3% to

16.2% and 1.1% to 54.1%, with average values of 4.0% and 15.8%, respectively. There was a strong linear correlation between STD and Range (slope = 3.5,  $R^2$  = 0.914). In around one third of the catchments (16 of 50), the uncertainties of  $C_{sub}$  were relatively small (STD 

Figure 6. Boxplot of the uncertainty metrics of  $C_{\text{sub}}$  produced by behavioral parameter sets in the selected 50 catchments

Changes in C<sub>sub</sub> uncertainty with varying behavioral thresholds were further analyzed by adjusting the difference between KGE<sub>opt</sub> and the KGE threshold from 0.005 to 0.05 in 0.005 increments (Figure 7). As expected, most metrics increased as the KGE threshold decreased, although some metrics showed non-monotonic extrema (e.g., the maximum RBias). Notably, uncertainty metrics could still be large even when the KGE threshold was set very close to KGE<sub>opt</sub>. For instance, in the Yongshun catchment (Figure 4b), Bias and Range reached 12.2%

and 38.8%, respectively, when the KGE threshold was set only 0.005 below  $KGE_{opt}$ . However, the 90th percentiles of Bias and Range were below 5% and 10%, respectively, when the KGE threshold was set 0.01 below  $KGE_{opt}$ , indicating that  $C_{sub}$  estimation is robust in most catchments if the threshold is set sufficiently high.

Figure 7. Changes in  $C_{\text{sub}}$  uncertainty metrics with changing behavioral KGE thresholds with an interval of 0.005

# 3.3 Influence factors of surface-subsurface runoff partitioning

To further understand the variability of surface—subsurface runoff partitioning and its uncertainties, we analyzed their correlations with the catchment attributes listed in Table 1. Considering the strong correlation between uncertainty metrics STD and Range, STD was not included in the correlation analysis for simplify. The correlation coefficients and the corresponding statistical significance are shown in Table 3. For the contribution of subsurface runoff, runoff ratio (QR) was the only attribute showing a statistically significant correlation (p<0.01) with C<sub>sub</sub>, indicating greater subsurface contribution in catchments with stronger runoff generation capacity. Notably, C<sub>sub</sub> exhibited a positive correlation with topographic slope (TS), which is consistent with the global isotope-based study by Jasechko et al. (2016) that found the young-water fraction (associated with surface-runoff contribution) to be negatively

correlated with topographic gradient, although the correlation in our study was not statistically significant (p = 0.26).

Table 3. Correlation coefficient among C<sub>sub</sub>, uncertainty metrics, and catchment attributes among 50 catchments. Bold values indicate p<0.05, with \* and \*\* denoting 0.01<p<0.05 and p<0.01, respectively.

|           | $C_{Sub}$ | Bias     | RBias    | Range   |
|-----------|-----------|----------|----------|---------|
| DRA       | 0.055     | 0.167    | 0.025    | 0.031   |
| MAR       | -0.256    | -0.387** | -0.289*  | -0.244  |
| MAQ       | 0.199     | -0.128   | -0.374** | 0.019   |
| QR        | 0.632**   | 0.434**  | -0.126   | 0.380** |
| TS        | 0.162     | 0.064    | -0.055   | 0.251   |
| $C_{Sub}$ | \         | 0.074    | -0.563** | 0.217   |

The uncertainty metrics were correlated with both climate conditions and runoff-generation capacity. Bias and RBias were both significantly negatively correlated with mean annual rainfall (MAR) (p<0.01 and p<0.05, respectively). RBias was also significantly negatively correlated with mean annual runoff (MAQ) (p<0.01). These results suggest that  $C_{\text{sub}}$  estimated from the optimal parameter set tended to be closer to the mean  $C_{\text{sub}}$  across all behavioral sets in wetter catchments (Figure 8). As expected, RBias was strongly negatively correlated with  $C_{\text{sub}}$  (r=-0.563), because  $C_{\text{sub}}$  appears in the denominator of the RBias calculation. QR was significantly positively correlated with Bias (r=0.434, p<0.01). The correlations between Range and the examined catchment attributes were similar to those for  $C_{\text{sub}}$  in both strength and statistical significance. QR was the only attribute significantly correlated with Range (r=0.380, p<0.01).

It is worth noting that QR was significantly correlated with both surface—subsurface runoff partitioning and its uncertainties (Figure 9b–d). We also examined the relationship between model sensitivity—represented by the number of behavioral parameter sets obtained from 50 pySOT calibrations—and QR, and found a strong, significant correlation (r=0.731, p<0.01; Figure 9a). This suggests the model was less sensitive (i.e., produced a larger number of behavioral parameter sets) in catchments with higher QR. The positive correlation between C<sub>sub</sub>

315316

317318

and QR likely reflects that a high QR implies a smaller fraction of rainfall lost to evapotranspiration, favoring larger subsurface water storage and lateral subsurface flow. Additionally, higher QR implies more runoff generation via multiple pathways, which may increase variability and consequently the uncertainty in  $C_{\text{sub}}$ .

**Figure 8.** Scatter diagram of (a) Bias and MAR, and (b) RBias and MAQ. Blue dashed lines represent the regression fit.

Figure 9. Scatter diagram of QR and (a) number of behavioral parameters, (b)  $C_{sub}$ , (c) Bias and (d) Range. Blue dashed lines represent the regression fit.

#### 4. Discussions

327328

346347

# 4.1 Implications on hydrological modelling in various conditions

This study questions the reasonableness of commonly used hydrological model calibration strategies. Results show that in some catchments model performance is insensitive to internal process representation — a very small change in an evaluation metric (for example, a 0.005 change in KGE) can correspond to very large differences in the estimated contribution of subsurface runoff. Similar phenomena were also reported for other runoff components, such as snowmelt and glacier melt runoff (He et al., 2019; Nan et al., 2025). In such cases, the optimal parameter set may not adequately represent the range of plausible hydrological behaviors, and analyses based on a single parameter set may therefore yield substantially biased estimates. Conversely, in a substantial proportion of catchments (approximately one-third in this study), uncertainties among behavioral parameter sets were relatively small, and the optimal parameter set can adequately represent the parameter sets that produced sufficiently high KGE. The correlation analysis provides potential guidance for choosing appropriate calibration strategies under different catchment conditions. The runoff ratio (QR) emerged as a representative index for a priori assessment of potential modeling uncertainties: catchments with higher QR tended to exhibit greater uncertainties. In such cases, a systematic evaluation of model uncertainty, such as adopting Generalized Likelihood Unvertainty Estimation (GLUE) framework, is recommended. Uncertainties can be further reduced by assimilating multiple datasets (e.g., Tong et al., 2021). Conversely, in catchments with lower QR the model was more sensitive to parameterization, so a single parameter set that yields the optimal performance metric is more likely to credibly represent the underlying hydrological processes. In those cases, calibration that targets a single optimal parameter set may be acceptable. Mean annual rainfall (MAR) was identified as a secondary representative index for a priori assessment of modeling uncertainties. MAR had a significant negative correlation with Bias (Figure 8a), and a negative (but not statistically significant) correlation with Range (r = -0.244, p = 0.087; Table 3). Meanwhile, the number of behavioral parameter sets was significantly negatively correlated with MAR (r = -0.379, p

These results indicate that modeling uncertainties tend to be lower in catchments with lower runoff generation capacity and wetter climates. However, it is difficult to derive a reliable equation to predict potential modeling uncertainty from catchment attributes, or to define a MAR/QR threshold above or below which a single optimal parameter set can be judged sufficiently credible. For example, among the five catchments with MAR>1500 mm and QR

Figure 10. The interrelation among the QR, MAR, and (a) Bias and (b) Range

#### 4.2 Limitations

This study used the contribution of subsurface runoff (C<sub>sub</sub>) estimated by the THREW model as a representative hydrological characteristic to analyze hydrological-model uncertainty and its influencing factors. The main limitation of this study lies in the reliability of the estimated C<sub>sub</sub> and in the representativeness of the THREW model. Because of deficiencies in observational data for multiple runoff components, it is difficult to directly validate the C<sub>sub</sub> estimates in this study, which consequently raises doubts about conclusions related to C<sub>sub</sub>. Nonetheless, some indirect evidence supports the C<sub>sub</sub> estimates to some extent. On the one hand, in our previous studies the C<sub>sub</sub> estimates produced by the THREW model in two typical catchments were independently validated by other methods, such as an end-member mixing model and a groundwater model (Nan et al., 2021; Nan et al., 2023; Yao et al., 2021; Li et al., 2020), demonstrating the THREW model's ability to simulate surface—subsurface runoff partitioning. On the other hand, the positive correlation between C<sub>sub</sub> and topographic slope and

the negative correlation between C<sub>sub</sub> and mean annual rainfall are consistent with patterns reported in previous studies (Jasechko et al., 2016; Nan et al., 2025), which provides additional support to the reasonableness of our results.

Another limitation is the use of a single THREW model, which raises questions about the generalizability of the conclusions. However, the structure and core equations of THREW model are broadly similar to those of many commonly used semi-distributed hydrological models, such as SWAT and HBV (Gassman et al., 2007; Lindström et al., 1997). Specifically, the basic simulation units of THREW are subzones of the representative elementary watershed (REW), defined by topography and land-surface type — analogous to the hydrological response unit (HRU) concept in other models. Hydrological processes are computed within each unit, including surface runoff, infiltration, subsurface flow, evapotranspiration, and changes in water storage. We therefore expect qualitatively similar findings across semi-distributed models, although specific quantitative outcomes may differ depending on parameterizations (Beck et al., 2016), data conditions (Seibert & Beven, 2009), process representations (Knoben et al.,

2020), spatiotemporal resolution (Zhang et al., 2025), and calibration strategy (Sun et al., 2020).

Future work could include multi-model comparisons or ensemble simulations to assess how

model structural differences affect the robustness of the conclusions.

## 5. Conclusion

This study used the semi-distributed THREW model to investigate the influence of parameter equifinality on runoff component partitioning. The model was implemented in 63 catchments in southeastern China with high-quality rainfall and streamflow records. Behavioral parameter sets were selected using a KGE threshold (KGE $_{opt}$ –0.05) to quantify uncertainty in estimates of the contribution of subsurface runoff ( $C_{sub}$ ). Correlation analyses were conducted among  $C_{sub}$ , uncertainty metrics, and catchment attributes to identify influence factors. Our main findings are as follows:

1. The THREW model showed generally good performance across the 63 catchments, with KGE $_{opt}$  ranging from 0.663 to 0.947 (mean = 0.846).  $C_{sub}$  varied widely among catchments, from 1.0% to 74.1% (mean = 31.7%).  $C_{sub}$  was lower than 50% in 84% of the catchments,

reflecting dominant surface runoff in the study area.

2. Small differences in KGE can produce substantial uncertainty in  $C_{sub}$ , and the magnitude of that uncertainty varied markedly among catchments. Bias (difference between the  $C_{sub}$  estimated using the optimal set and the average across all behavioral parameter sets) ranged from 0.1% to 19.8% (mean = 2.7%), and Range (difference between the maximum and minimum  $C_{sub}$  among behavioral parameter sets) ranged from 1.1% to 54.1% (mean = 15.8%). In catchments with larger uncertainty metrics, the calibration strategy focusing on a single optimal parameter set is likely to result in misestimation of internal hydrological processes.

3. Runoff ratio (QR) was identified as the primary catchment attribute associated with both surface-subsurface runoff partitioning and its uncertainty. QR was significantly correlated with C<sub>sub</sub>, Bias and Range (p

publication.

# Data and Code Availability

- Datasets used for model setup are available at: Yamazaki et al. (2019), Muñoz Sabater,
- (2019), Dai et al. (2019), Myneni et al. (2021) and Didan (2021). The basic information of the
- catchment set and the model code of a typical catchment are available at Zenodo:
- https://zenodo.org/records/17089365.

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
