# Peer review of "Larger hydrological simulation uncertainties where runoff generation capacity is high: insights from 63 catchments in southeastern China Yi Nan1\*, Fuqiang Tian1,2, Mahmut Tudaji1,2 1. State Key Laboratory of Hydroscience and Engineering, Tsinghua University, Beijing"

_EGUsphere, 2025_

## Author Comment (AC1)

**Response to Reviewer #1**

The study quantified the effect of parametric uncertainty on the resulting runoff partitioning using 50 behavioral datasets of the Tsinghua Hydrological model in more than 60 catchments in the Yangtze River Basin. It shows that in catchments with larger runoff ratio, the parametric uncertainty is higher resulting in higher variability of simulated runoff components.

Although I find the analysis of relationship between parametric uncertainty and their possible catchments attributes interesting in general, the investigation performed in this study is rather incomplete and poorly motivated. The model appears to be unvalidated. Moreover, a rather small sample of parameters were considered for the analysis, compromising the reliability of the results. No rationale for selecting possible catchment controls is provided, while the results are based on simulation of a single model and single objective function, making the generalizability rather difficult. Please find my detailed comments below.

**Response:**

Thank you very much for your constructive comments. The points you raise indeed represent important limitations of our study. Some of these can be addressed by conducting supplementary experiments or clarified in the text, while others may be difficult to address adequately within the current scope of this work. Please find our detailed responses to the general and minor comments below.

General comments

1. Motivation: The motivation of the study as mentioned in Line 68-70 is to provide an a priori guidance on whether to only select single best calibration parameter or instead choose multiple behavioral parameter sets. I rather disagree with such premise. Even if some studies still use a single best parameter set, they are simply ignoring parametric uncertainty altogether, while there is a large body of studies including those referenced by the authors that show that using multiple parameter sets is essential for accounting for and communicating uncertainty. I do find the idea of knowing a priori how much parametric uncertainty one might expect useful for the sake of understanding and improving model limitations and to lower the computational costs of running a large number of simulations.

However, especially given rather inconclusive results of this study, using single parameter set would in any case mean ignoring the uncertainty. See also my detailed comments.

**Response:**

Thank you for the comment and we agree that our original premise regarding "guidance" for calibration strategy was indeed somehow overstated. This study mainly focuses on understanding the influence factors of the uncertainty in runoff component partitioning simulations, which is also valuable for understanding hydrological model behavior and limitations. Given the current inconclusive results, it is difficult to provide a "guidance" for the choice of calibration strategies. We will rephrase the motivation of the study in the revised manuscript.

2. Lack of Validation: It seems that there is no validation of model performance provided in this study. The method section describes only the calibration approach. Moreover, it seems that the total length of observations used for calibrations is about 2 years, which is too short to result in reliable parameter identification. Without the validation, there is no proof of parameter validity for an uncalibrated period.

**Response:**

Thank you for the comment. We acknowledge the shortage of data length and the lack of validation. However, we need to clarify that publicly available streamflow records are limited for Chinese basins, and access often depends on collaboration with data-providing agencies (Chen et al., 2025, https://doi.org/10.1038/s41561-025-01833-x; Lin et al., 2023, https://doi.org/10.1038/s44221-023-00039-y). The dataset used in this study covering 63 hydrological stations was obtained through such collaborations and represents a comparatively rare multi-basin observational resource. Even though it covers only two years, obtaining concurrent records from 63 stations was challenging.

Given the data scarcity, we utilized the full 2-year period for calibration to take full advantage of the data. The model performance is expected to be rather stable due to the short period, i.e., the model is unlikely to perform well in one year but extremely poorly in another year because of the similar conditions. However, to address the concern about model stability and validity, we will conduct a supplementary experiment. In specify, we will select representative

catchments and split the data into a calibration period (Year 1) and a validation period (Year 2), to analyze the correlation between the KGE performance in the calibration and validation periods. We will include these results in the Supplementary Material and discuss them in the Methods section to justify the robustness of our approach despite the data limitations.

3. Parameter sampling: Given that the focus of this study is to investigate parameter uncertainty, 50 parameter sets seem like a very low number of samples. A much more comprehensive sampling is needed to prove that lower number of behavioral sets is indeed not simply an artefact of small sampling.

**Response:**

Thank you for the comment. We apologize for the lack of clarity regarding the parameter sampling strategy. The number 50 is the final selected behavioral parameter sets, not the total number of samples evaluated. Our calibration utilizes the pySOT algorithm, which involves a two-layer loop structure: (1) Inner loop: In each pySOT run, the model is evaluated 3,000 times to find a local optimum. (2) Outer loop: We repeated the pySOT run 50 times to avoid being trapped in local optima. Therefore, the total number of parameter evaluations is actually 150,000 (3,000*50). The 50 sets represent the local optimal solutions derived from this extensive sampling. We will clarify the sampling strategy in the Methods section to demonstrate that the parameter space has been adequately explored.

4. Single calibration strategy and single model: While the study aims to provide generalizable results on how to a priori decide calibration strategy, it only examines one single objective function for the calibration (i.e., KGE) and one single conceptual model. In the Limitation section the authors highlight the latter as the limitation themselves. As hydrological models of even rather similar structures are known to behave quite differently in how they partition the fluxes (Merz et al., 2022 https://doi.org/10.1175/BAMS-D-21-0284.1; van Kempen et al., 2020 https://doi.org/10.5194/nhess-21-961-2021), several model structures must be analyzed to confirm reliability of the suggestions presented here. Similarly, given high effect of the objective function on the resulting parameter sets, a comprehensive set of experiments with various objective functions is needed to evaluate the generalizability of

the findings.

**Response:**

Thank you for your comment. We agree that selection of models and objective functions influence the results. However, given the vast diversity of hydrological models and the substantial workload involved in establishing and calibrating models for 63 catchments, it is challenging to implement a comprehensive multi-model comparison within the scope of the current study. Similarly, regarding the objective function, we selected KGE as a representative metric because it is a comprehensive metric that balances bias, variability, and correlation. In the revised manuscript, we will explicitly acknowledge this as a limitation, and that future work should extend this analysis to other model structures and objective functions to test generalizability.

5. Potential controls of uncertainty: The manuscript does not provide the rationale for selecting a few examined catchment attributes as potential controls of parameter uncertainty. This is necessary to understand why the examined attributes were selected in a first place and if there any other potential properties that can be used to explain observed parameter uncertainty.

**Response:**

Thank you for the comment. At this point, there was no formal selection rationale applied in this exploratory study. We selected a small set of basic, commonly used, physically meaningful catchment attributes that either influence the rainfall-runoff processes (e.g., climate and topography conditions) or reflect the fundamental characteristics of these processes (e.g., runoff volume and runoff ratio). We will add a short note in the Methods to make this explicit and include a brief limitation in the Discussion clarifying that other properties may also affect parameter uncertainty and remain for future work.

Specific comments

Line 18, 111 and elsewhere: It is not quite clear what is meant by "high-quality rainfall" here? It is also was not clarified later in the Methods part. It would be much more instructive to specify temporal resolution, length of observations or density of the observations to highlight a

particular aspect of data quality. Please revise.

**Response:**

The term "high-quality" mainly referred to the high temporal resolution (< 1h) and integrity (low missing rate) of the data. We will remove the vague term and instead explicitly describe the data attributes in the Methods section.

Line 23: Given the theme of this paper, I do not think that the model performance is relevant for the abstract. Consider omitting this.

**Response:**

We agree that the model performance is not so relevant for the abstract, but we believe that good model performance is a necessary basic for the subsequent uncertainty analysis. Consequently, we will simplify this part in the abstract to only brief state that the model performs adequately well.

Line 25-26: At this point, this statement is not clear, because it is not yet clarified how the uncertainty is computed. Please clarify and revise.

**Response:**

We will replace the vague phrasing with more precise terms such as "range" to clearly define how the uncertainty was quantified.

Line 34-36: This recommendation is rather general. It would be helpful to specifically highlight what sort of guidance this study can provide for the choice of the calibration strategy.

**Response:**

The specific recommendation is actually the third point of result about the relation between model uncertainty and catchment attributes. However, as pointed out by the reviewer and acknowledge in the response to the general comments, the current results are not conclusive enough for providing an operational guidance. Consequently, we will remove the statement regarding guidance from the Abstract and focus on the results.

Line 49-51: This recent review of Wagner et al., 2025 (https://doi.org/10.1002/wat2.70018)

might be worth mentioning here.

**Response:**

We will include this reference in the introduction as suggested.

Line 53: I disagree that these are two different calibration approaches. Considering multiple behavioral parameter sets vs taking one best single parameter set stands for "accounting for parametric uncertainty" vs "ignoring it". While most of the current modeling studies account for parametric uncertainty using various approaches, using a single best parameter set remains a poor practice that unfortunately still persists in some studies. Yet, this does not make it a distinct calibration strategy. Please revise.

**Response:**

We agree that adopting multiple behavioral parameter sets and one best single parameter set are actually not two different calibration approaches, but two different treatments to model parameter uncertainty. We will revise the statement in the introduction. However, we think that the necessity of a detailed uncertainty analysis often depends on the specific research objectives and questions. While parameter uncertainty is undoubtedly important, it may not always be the primary focus of every hydrological study, and we cannot expect every single study to address all aspects of modeling comprehensively. Even in our current work, which prioritizes parameter uncertainty, we specifically focus on the uncertainty of runoff partitioning. Therefore, we will modify the text to reflect that using a single parameter set is a simplification often driven by specific research goals, rather than presenting it as a competing strategy to uncertainty analysis.

Line 57-59: Given the relevance of the behavioral parameter sampling, this part seems rather short to me and incomplete, especially in terms of the references mentioned. It seems that most of seminal works of Keith Beven on this topic are overlooked here. Please add.

**Response:**

We will add the relevant citations by Keith Beven and others.

Line 63-64: As I mentioned in my comment above, even if some studies still use a single parameter set, there is a consensus that in order to represent parametric uncertainty multiple

parameter sets should be considered as the study referenced here also emphasizes.

**Response:**

We will adjust the statement.

Line 99 and elsewhere: The term "average" is ambiguous. Please specify if this is mean or median.

**Response:**

It refers to the arithmetic mean. We will specify "mean" throughout the manuscript.

Line 101: Please clarify here what is meant here by runoff ratio and how it was computed in this study.

**Response:**

The runoff ratio is calculated as the ratio of total discharge to total precipitation. We will clarify this definition in the revised Methods section.

Figure 1: This figure would be more instructive if it would also display precipitation or runoff ratios that are used as explanatory controls of the uncertainty in this study.

**Response:**

We will display these basin attributes in Figure 1 in the revised manuscript.

Line 112-113: This is a rather misleading way report missing values. Please specify the study period and the portion of missing data across all catchments.

**Response:**

The study period for all the 63 basins is consistently 2014.1.1-2015.12.31. We will report the portion of missing data in the revised manuscript.

Line 115: Please quantify this by specifying how much water level data vs discharge data is available.

**Response:**

We will add the specific number of water level and discharge data in the revised manuscript.

Line 116-119: Please specify how these relationships were built. Was it a linear relationship?

**Response:**

Yes. The linear relationship was used. We will clarify this.

Line 128-132: It is not clear how this information was used. Is it needed for model input? This has to be clarified.

**Response:**

Yes, these datasets were used as model inputs. We will add few sentences to clarify this in the manuscript.

Line 149: Even if the model was used in the previous studies key equation and especially parameters have to be specified. Parameters can easily be added near the corresponding compartment in Figure 2. Please add.

**Response:**

We will update Figure 2 to illustrate each parameter controls which process. However, as the THREW model is a complex distributed model containing several subregions, it is difficult to say which equations are the most important. Consequently, we will provide brief description in the manuscript and cite the papers about the model development for the detailed mathematical formulations.

Section 2.3: This section must provide a complete information on the length of calibration period and validation period.

**Response:**

As clarified in the response to general comment, we utilize all the observation streamflow data to calibrate the model considering the short length of data series.

Table 2: Are these all the parameters of the model? Please clarify.

**Response:**

These are the parameter determined by calibration. There are also some parameters determined

by related dataset (e.g. soil property parameters) or fixed due to small influence on model performance. We will clarify this in the table caption and main text.

Line 172: It is not clear what is meant by the "optimal KGE", the highest one?

**Response:**

Yes, it means the highest KGE. We will rephrase for clarity.

Line 179: Even if the river bed might be impervious, this is not a primary reason for surface runoff. For perennial rivers as here, rainfall falling on the channel surface is much more likely to flow horizontally along the channel than vertically towards river bed. Please revise.

**Response:**

Yes, the original expression was inaccurate. We intended to refer to the water surface of the river channel. We will delete "impermeable areas" and directly specify "river channel water surface" avoid confusion.

Section 3.1: It is not clear of the reported performance corresponds to the calibration or to the validation method. Please clarify.

**Response:**

As clarified in the response to general comment, we utilize all the observation streamflow data to calibrate the model considering the short length of data series. We will clarify it here.

Figure 8: Please add explanation of clusters in the caption. Please also clarify what is meant here by maximum rainfall. Event? Rate? Volume?

**Response:**

It is mean annual rainfall as illustrated in Table 1. We will also add explanation of these terms in the caption.

Line 206: Please clarify what makes these two catchments typical and why for another example later another set of two catchments were selected. Please avoid subjective choice of catchments for examples and present the results for all study catchments.

**Response:**

Visualizing detailed time-series for all 63 catchments in the main text is not feasible, so we only presented the results for some catchments. Different typical catchments were selected for Figure 4 and Figure 5 to represent different aspects of behaviors. Figure 4 shows the model performance, so the catchments with highest and lowest KGE were selected as typical catchments. Figure 5 illustrates the relation between KGE and the contribution of subsurface runoff ($C_{sub}$), so the catchments with high and low sensitivity of KGE to $C_{sub}$ were selected. We will clarify this selection logic in the text. Additionally, although presenting results for all catchment in main text is not feasible, we will provide the data producing Figure 4 and Figure 5 for all catchments in the Supplementary Information to ensure completeness.

Figure 3: This figure could be more efficiently presented as a boxplot or a violin plot.

**Response:**

We will redraw the Figure 3 accordingly.

Line 216 and elsewhere: Please avoid term "significantly" if no statistical test was used.

**Response:**

We will check through the manuscript and remove all such terms.

Line 260: Please be more specific on which catchments are these.

**Response:**

We will provide the information of the specific catchment.

Figure 6 and all other figures: Please explain all the terms and acronyms from the figure in the caption.

**Response:**

We will explain the terms and acronyms in the caption of all figures.

Table 3 and elsewhere: Please specify type of correlation used here.

**Response:**

The "correlation coefficient" reported here and elsewhere is the Pearson correlation coefficient (r), which measures the linear correlation between the two variables. We will clarify it in the revised manuscript.

Line 306: I am not sure if the term "model sensitivity" is suitable here. Given that "parameter sensitivity" is a rather established term, I would avoid using it in a different context.

**Response:**

We will replace "model sensitivity" with a more accurate term to avoid confusion with the established term.

Line 338: It is not clear what is meant here by the assimilation of the datasets. Please clarify.

**Response:**

It means use multiple datasets to constrain uncertainty and improve model performance. As we are not referring to a specific practice, so a rather general term is used here.

Line 367-371: Please specify if these studies were in the same study region/ catchments.

**Response:**

Those studies were not in the same region with this manuscript. We will clarify it in the revise manuscript.

---

## Author Comment (AC2)

**Summary**

The main goal of the paper is to analyze if predictors can be found that can indicate if calibrating a particular catchment only once will be sufficient to accurately estimate internal hydrological processes and variables (in this case the fraction subsurface flow), or that calculating an ensemble mean over multiple calibrations may be necessary for an accurate estimation because of parameter equifinality.

The authors calibrated the THREW model 50 times for each of the 63 catchments that were used in this study. The catchments were calibrated on streamflow using KGE as the objective function. A threshold value was applied to select only those parameter sets that performed reasonably well in terms of KGE. Four different measures were defined to describe the uncertainty in the estimated fraction subsurface flow resulting from simulations with the calibrated parameter sets. Linking the uncertainty measures to catchment characteristics, the authors found that for catchments with higher runoff ratios the estimation of the fraction subsurface flow was less certain.

The paper is well written and easy to follow, the language is clear, and the general topic of uncertainties in calibration strategies is relevant and interesting. However, I have a few concerns with regard to the methodology that is used within this study, which I will outline below, together with some other major and minor feedback.

**Response:**

Thank you for your positive evaluation on our manuscript and your constructive suggestions. We will revise the paper according to your comments. Regarding the use of behavioral parameter sets for analysis, after careful discussion among the authors, we decided to retain this approach as it aligns well with the scope and aims of our study. Please see our detailed response to the relevant comment below.

**Major comments:**

The authors compared two calibration strategies: calibrating the model only once and calibrating the model 50 times. They defined two measures (Bias and RBias) to analyze how representative the contribution of subsurface runoff (Csub) estimated by the parameter set with the highest KGE is. However, calibrating the model only once can results in any

of the 50 parameter sets with the same likelihood. There is no guarantee (and one cannot know) if the calibration resulted in the parameter set with the highest KGE value. Therefore, measures analyzing the representativeness of the best parameter set (in terms of KGE) seem to me to be not meaningful for the purpose of this study. The 'Bias' measure, for example, only shows to which extent Csub modeled by the best parameter set corresponds to the mean Csub of all calibrations, but it doesn't say anything at all about how wrong or inaccurate Csub can possibly be for a single calibration.

The authors calibrated the model for each of the catchments 50 times. They then selected those parameter sets whose KGE exceeded a certain threshold, the so-called behavioral parameter sets, and used them for further analyses [Line 171-172]. I think this selection is problematic. Any of the removed parameter sets could be the result of a single calibration. Figure 5b clearly shows the possible consequences of excluding parameter sets from further analyses for the Shuikou catchment. For this catchment, all behavioral parameter sets have a Csub value roughly between 25 and 32 (a rather small spread), but a single calibration could result in a Csub value of at least up to 55. I understand that if a calibrated parameter set doesn't model the discharge very well, one may have less trust in the modeled Csub. It's unfortunate that the optimization strategy was not very effective, given the often small number of behavioral parameter sets. But that doesn't mean that one can simply ignore the fact that a single calibration could result in any of the non-behavioral parameter sets.

I think it is important to zoom in a bit on the possible consequences of using a performance threshold for the results and/or conclusions of the paper. Even though the authors mention that it is difficult to define a QR threshold below which a single optimal parameter set can be judged sufficiently credible [Line 351-353], the paper implicitly concludes that the smaller the QR ratio, the smaller the uncertainty measures, and therefore the more likely that one calibration may be sufficient to accurately estimate Csub. However, a strong, significant correlation was found between QR and the number of behavioral parameter sets [Line 305-307]. So, the lower the QR ratio, the more parameter sets were removed from the analysis due to the calibration algorithm struggling to find a good fit. In other words, for those catchments for which one calibration might potentially be sufficient according to

the analyses within this study, chances are rather high to end up with a parameter set that does not even calibrate the discharge well.

**Response:**

We thank the reviewer for this critical and interesting perspective. The reviewer raises a reasonable concern that a single random calibration does not guarantee finding behavioral parameter sets, and therefore, analysis based on behavioral parameter sets might seem not meaningful.

We have seriously considered the reviewer's suggestion to modify the analysis. However, after re-evaluating our research objectives, we believe that the current analytical framework remains reasonable for our specific study goals. Consequently, we do not intend to change the analysis strategy (i.e., still based on behavioral parameter sets), but we will improve the clarity of our methodology and objective, and add a discussion on limitations.

The reason for retaining the current analysis can be summarized as fourfold:

1. **The research objective is not to assess the performance of a single calibration runoff.** The primary goal of this study is not to assess whether a single calibration run is reliable, which is actually a question of algorithmic stability. Rather, we aim to answer: "Can the specific optimal parameter set (the one with the highest KGE) sufficiently represent the runoff component partitioning behavior derived from the ensemble of behavioral parameter sets?" In other words, we are comparing the representativeness of the "best model" strategy against the "ensemble" strategy, not the probability of a random calibration run's success.

2. **Behavioral model performance is the precondition of a reliable analysis on hydrological processes.** As the reviewer noted, non-behavioral parameter sets are inherently unreliable. In hydrological modeling, if a parameter set fails to reproduce discharge accurately, it is expected to be unable to simulate internal hydrological processes well, and is typically discarded. Our analysis focuses on the "behavioral" space because comparing the "best" set against "unreliable/poor" sets would not yield physically meaningful conclusions.

3. **Analysis based on a single optimal parameter set doesn't indicate to calibrate the model only once.** In our study, "using a single parameter set" refers to the practical workflow where a modeler selects the single best result for subsequent analysis (regardless of how many iterations it took to find it). In practice, even when analyzing based on the single optimal parameter set, it is generally obtained by a large number of calibration runs. We are assessing the uncertainty introduced by relying on this selected optimal set versus using the full behavioral ensemble.

4. **Analyzing the randomness of a single calibration runoff will introduce additional complexities related to the optimization algorithm used.** For instance, in our implementation of the NSGA-II algorithm, the optimal parameter set shows high stability and is highly reproducible when run multiple times under the same settings (e.g., calibrated parameter ranges, time of model runs). Focusing on the stochastic nature of the search process would shift the paper's focus toward algorithmic, which is beyond the scope of this hydrological study.

Although we will maintain the current data analysis, we acknowledge that our terminology led to this misunderstanding. Therefore, we plan to make the following revisions:

- We will explicitly state that our comparison is between the identified optimal parameter set and the behavioral ensemble, clarifying that we are not evaluating the "first random run" of an optimization algorithm.

- We will add a discussion in the Limitations section to address the reviewer's concern. We will acknowledge that in practice, if the number of calibration trials is insufficient, one might fail to identify a behavioral set (the "optimization failure" scenario). We will clarify that our study assumes a behavioral solution has been found, and that the uncertainty of "failing to calibrate" is a separate issue from the parameter equifinality discussed here.

**Specific questions / feedback:**

- Line 88-90: "(1) to quantify the uncertainty in the contribution of subsurface runoff (Csub) resulting from small changes in model performance metric (KGE)" -> As outlined earlier, for the purpose of this study, this should to my opinion have been e.g.: '*to quantify the uncertainty in the contribution of subsurface runoff (Csub) when calibrating a hydrological model multiple times*'.

**Response:**

We understand the reviewer's perspective. As explained in our response to the major comment, however, the focus of our study is not on the calibration process itself, but rather on the subsequent analysis using the calibrated parameters. Our goal is to assess how much uncertainty in Csub estimation arises when the analysis is based on a single optimal parameter set, using the ensemble of behavioral parameter sets as a benchmark. Since the estimation of runoff components is only meaningful when the model performs adequately well, we believe the original statement is appropriate in the context of our study.

- Line 159-160: "the optimization stopped when the objective converged or the number of model runs reached a threshold" -> Which of the 2 happened for the different calibrations within this study? Considering the often large number of non-behavioral parameter sets, it would be valuable to know if the calibration got stuck in a local optimum or reached the maximum number of model runs. In case of the latter, the number of model runs during each calibration might not have been set high enough.

**Response:**

In most cases, the calibration runs stopped due to convergence of the objective function rather than reaching the maximum number of model runs. In practice, we found that the limit of 3000 model runs per calibration was usually sufficient for the pySOT algorithm to reach a local optimum. Unfortunately, we did not retain detailed iteration records for each calibration and basin, so we cannot quantify exactly how often each of the two stopping criteria occurred. Nevertheless, we will clarify in the revised manuscript that, in most cases, the optimization terminated due to objective function convergence rather than the run limit.

- Line 204-205: KGE values for the best parameter sets are given, but how are the values for the other 49 parameter sets (range and/or distribution)?

**Response:**

We will provide the KGE range of 50 parameter sets in each basin in the Supplementary Information. However, since this section aims to summarize the model performance, only the best KGE values are reported in the main text.

- Line 219-221: "Considering the random generation of initial parameter sets within each pySOT running, the number of behavioral parameter sets serve as a partial indicator of model sensitivity." -> I don't agree with this statement. PySOT is an optimization algorithm, and the number of behavioral parameter sets tells more about how easy or difficult it is for the algorithm to find the optimum, i.e. about the 'smoothness' and the exact structure of the parameter space.

**Response:**

Thank you for pointing this out. We acknowledge that the phrase "model sensitivity" was not used accurately in this context. The number of behavioral parameter sets is indeed more indicative of the structure of the parameter space and the ease with which the optimization algorithm finds good solutions, rather than sensitivity in the strict sense. We will revise the term here.

Nonetheless, we would like to further discuss this point with the reviewer. As the reviewer noted, the number of behavioral parameter sets reflects the smoothness and structure of the parameter space. In other words, it reflects the relationship between model parameters and performance. Could this not also be interpreted as a reflection of the sensitivity of model performance to parameter changes?

- Line 271-273: "the 90th percentiles of Bias and Range were below 5% and 10%, respectively, when the KGE threshold was set 0.01 below KGEopt, indicating that Csub estimation is robust in most catchments if the threshold is set sufficiently high" -> I do not fully understand the reasoning here. Why exactly do the percentiles need to be low? If it is about robustness, one can set the threshold to KGEopt. Then there will be only one parameter set left, resulting in an 'excellent robustness' of Csub, with Bias, RBias, STD and Range all equal to 0...

**Response:**

We apologize for the confusion. In this context, the "90th percentile of Bias" refers to a statistic across the 50 catchments in our study: it means that 90% of the catchments have a Bias value below 5% when the KGE threshold is set to 0.01 below KGEopt. The metric that reflects the robustness of the Csub estimation is the Bias and Range themselves, rather than the "90th percentile" value.

However, as the reviewer noted, if the KGE threshold for defining behavioral sets is set sufficiently close to the optimum, the uncertainty in Csub estimation will always be low. What we aim to emphasize here is that even when the threshold is set just 0.01 below the optimum, the uncertainty can still be high in some catchments, as indicated by the spread in uncertainty metrics (the light color band in Figure 7). We will rephrase this explanation in the revision to avoid giving the impression that our intention was to emphasize the robustness.

- Figure 9: Is there a correlation between the number of behavioral parameter sets and the uncertainty measures (Bias and Range)? And could it be that the correlation between the uncertainty measures and QR is an artifact of the fact that not all parameter sets were used for the calculation of the uncertainty measures?

**Response:**

We have analyzed the relationship between the number of behavioral parameter sets and the uncertainty metrics. The number of behavioral sets does show a significant correlation with the uncertainty metrics, comparable to those between the runoff ratio (QR) and the corresponding metrics: specifically, the correlation between the number of behavioral sets and Bias is $r = 0.44$ ($p < 0.01$), and with Range is $r = 0.31$ ($p = 0.03$). We will add these results to the revised manuscript.

We acknowledge that restricting the analysis to only behavioral parameter sets may lead to these correlations. However, we do not consider this effect to be an "artifact", because excluding non-behavioral parameter sets is a reasonable and commonly accepted practice. As the reviewer noted, good model performance is a prerequisite for reliable analysis of internal processes such as Csub estimation. Therefore, applying a KGE threshold to select behavioral sets for the uncertainty analysis is, in our view, appropriate. Otherwise,

including a large number of poorly performing parameter sets would likely result in extremely large biases and uncertainties, making the analysis less meaningful.

- Line 331-332: "the optimal parameter set can adequately represent the parameter sets that produced sufficiently high KGE" -> But how can one know what is 'sufficiently high'?

**Response:**

In this study, we implicitly assume that, after 50 calibration runs (exploring tens of thousands of model evaluations), the global optimal parameter set for each catchment has been found. Consequently, "sufficiently high" KGE refers to those parameter sets whose KGE values are very close to the optimal KGE of that catchment (i.e., the behavioral parameter sets). We will clarify this more explicitly in the revised manuscript.

- Line 350-353: "However, it is difficult to derive a reliable equation to predict potential modeling uncertainty from catchment attributes, or to define a MAR/QR threshold above or below which a single optimal parameter set can be judged sufficiently credible". Let's assume that such an equation and/or threshold could be defined. Since for studies involving multiple catchments it is common practice to treat all catchments in an identical way in order to be able to make comparisons and draw overall conclusions, calibrating some of the catchments only once, and others multiple times may complicate the analyses and conclusions. Can the authors elaborate a bit on which studies could benefit from an equation or threshold to decide if a single optimal parameter may be sufficient, and how they would apply it in practice?

**Response:**

Thank you for this interesting and important question. We believe our findings could benefit two types of studies in particular:

1. Catchment characteristics analysis based on hydrological model: For studies that use hydrological models to analyze catchment characteristics across many basins (e.g., the contribution of subsurface runoff, Csub, as in our work), our conclusions could serve as a qualitative reference to flag catchments where larger uncertainty or bias is expected if only the optimal set is used for analysis. For instance, if a regional dataset of Csub is

generated using hydrological modeling, our findings suggest extra caution is warranted for catchments with low runoff ratios and high variability.

2. General hydrological modeling studies: Our findings may also serve as a reference for determining whether parameter uncertainty needs to be a primary focus in a given modeling study. Hydrological models are used to explore a wide range of scientific questions, often involving complex processes and multiple sources of uncertainty. In many cases, it is difficult or impractical to address all uncertainties comprehensively within a single study. Our results suggest that, in catchments with relatively high precipitation and low runoff ratios, the uncertainty introduced by parameter equifinality may be relatively minor. Therefore, in such cases, researchers might reasonably shift their focus toward other sources of uncertainty or research priorities. We would like to emphasize again that this does not imply that only a single calibration run should be performed. Rather, it means that once a well-calibrated optimal parameter set has been obtained, analysis based on this set alone may be sufficiently reliable, and ensemble-based analysis may not be necessary.

- Line 363-365: "Because of deficiencies in observational data for multiple runoff components, it is difficult to directly validate the Csub estimates in this study" -> Although validating the Csub estimates is indeed difficult, I would suggest to include a map with the catchments color coded by Csub. Spatial patterns of Csub in combination with expert knowledge about the characteristics and behavior of the different catchments might be of help to assess the likelihood of correct Csub values.

**Response:**

Thank you for your suggestion. We will add a map of Csub in each catchment in the revised manuscript.

- Line 368-369: "such as an end-member mixing model and a groundwater model" -> Can you elaborate a bit more on these models, in particular to which extent modeling was based on field measurements/data.

**Response:**

We will add a description of these two types of models. Briefly, the end-member mixing model relies on extensive tracer (e.g., isotope) measurement data, and the contributions of

different water sources are determined based on mass balance. Groundwater models focus specifically on subsurface flow processes, with detailed representations that differ from those in typical rainfall-runoff models. These two types of models are fundamentally different from our hydrological model. In the revised manuscript, we will clarify that in two other catchments, the groundwater contributions to streamflow estimated by our hydrological model were found to be similar to the results obtained using these different modeling approaches. This consistency across independent methods provides validation for our Csub estimates and increases our confidence in the reliability of our model results.

- Line 371-373: "the positive correlation between Csub and topographic slope and the negative correlation between Csub and mean annual rainfall are consistent with patterns reported in previous studies" -> Both correlations are not significant, and the correlation direction alone is not a very strong similarity.

**Response:**

Yes, this statement is indeed a little farfetched. We will delete this sentence in the revised manuscript.

**Minor comments:**

- Line 22-25: Are the KGE and Csub statistics for 63 catchments (as mentioned here) or for the 50 catchments that have at least 10 behavioral parameter sets (as mentioned in the results section)?

**Response:**

We made a mistake here. The statistics refer to the 50 catchments that have at least 10 behavioral parameter sets. We will correct this in the revised manuscript.

- Line 47: "Parameter calibration is a necessary step in developing hydrological models" -> This depends on the type of model.

**Response:**

Yes, you are correct. The statement here is inaccurate. We will adjust it and mention the model type in the revised manuscript.

- Line 63-64: "single–optimal-parameter approaches remain the default in many applications" -> Refs are 12-15 years old. Please support the statement with some newer publications.

**Response:**

We will add some recent publications conducting analysis based on the single optimal parameter set.

- Line 110-111: "Considering the variable quality of the raw data (mainly completeness and temporal resolution)" -> What do you exactly mean by the temporal resolution being of 'variable quality' other than having missing data (which falls under 'completeness')?

**Response:**

We apologize for the unclear phrasing. Apart from the difference in temporal resolution caused by missing data, the temporal resolution of the raw data also differs among catchments (ranging from 5min to >1day). Meanwhile, the completeness also varied. Considering both aspects, we set the average temporal resolution of 3650s, to select the catchments whose raw resolution is not less than 1h and a good completeness. We will clarify this in the revised manuscript.

- Line 113: 'Exceeded' or 'was below 3650'?

**Response:**

We apologize for the unclear phrasing. The average time interval of the data is below 3,650 seconds, but since "higher resolution" typically refers to shorter time intervals, our original wording might cause confusion. We will rephrase this sentence to explicitly refer to the data's time interval to avoid confusion.

- Line 121-124: Were lapse rates used?

**Response:**

Lapse rates were not used.

- Line 128: "Soil parameters" -> which parameters are these?

**Response:**

The soil parameters include soil properties such as saturated hydraulic conductivity, porosity, field capacity, etc. We will list the specific soil parameters in the revised text for clarity.

- Figure 2 (and the corresponding model description in the text):

o Use the same terminology consistently (e.g. u-zone, b-zone, n-zone etcetera).

o Leave out the components that are not used in this study (snow, glacier)

**Response:**

We will revise Figure 2 accordingly.

- Line 149: "in our previous publications" -> Include refs.

**Response:**

We will add the related references.

- Table 2: Where are the parameters exactly used within the model and what is their exact function? If possible integrate the parameters into Figure 2.

**Response:**

We will add the parameters to Figure 2 to illustrate which process each parameter influences.

- Table 3: Include the meaning of the abbreviations in the caption.

**Response:**

We will add the meaning of abbreviations in the captions in the revised manuscript.

- Line 337-338: "Generalized Likelihood Unvertainty Estimation (GLUE) framework" -> Add ref.

**Response:**

We will add related references in the revised manuscript.